# Nurses’ Views on the Use of Physical Restraints in Intensive Care: A Qualitative Study

**DOI:** 10.3390/ijerph18189646

**Published:** 2021-09-13

**Authors:** Federica Canzan, Elisabetta Mezzalira, Giorgio Solato, Luigina Mortari, Anna Brugnolli, Luisa Saiani, Martina Debiasi, Elisa Ambrosi

**Affiliations:** 1Department of Diagnostics and Public Health, University of Verona, Strada Le Grazie 8, 37134 Verona, Italy; Elisabetta.mezzalira@univr.it (E.M.); luisa.saiani@univr.it (L.S.); elisa.ambrosi_01@univr.it (E.A.); 2Cardiac Surgery Unit, San Bortolo Hospital, Viale Ferdinando Rodolfi, 37, 36100 Vicenza, Italy; giorgio.solato@gmail.com; 3Department of Human Sciences, University of Verona, Lungadige Porta Vittoria, 17, 37129 Verona, Italy; luigina.mortari@univr.it; 4Centre of Higher Education for Health Sciences, Azienda Provinciale per i Servizi Sanitari, Via Briamasco 2, 38121 Trento, Italy; anna.brugnolli@apss.tn.it (A.B.); martina.debiasi@univr.it (M.D.)

**Keywords:** intensive care, physical restraints, nurse, qualitative research

## Abstract

Despite the worldwide promotion of a “restraint-free” model of care due to the questionable ethical and legal issues and the many adverse physical and psychosocial effects of physical restraints, their use remains relatively high, especially in the intensive care setting. Therefore, the aim of the present study was to explore the experiences of nurses using physical restraints in the intensive care setting. Semi-structured interviews with 20 nurses working in intensive care units for at least three years, were conducted, recorded, and transcribed verbatim. Then, the transcripts were analyzed according to the qualitative descriptive approach by Sandelowsky and Barroso (2002). Six main themes emerged: (1) definition of restraint, (2) who decides to restrain? (3) reasons behind the restraint use, (4) physical restraint used as the last option (5) family involvement, (6) nurses’ feelings about restraint. Physical restraint evokes different thoughts and feelings. Nurses, which are the professionals most present at the patient’s bedside, have been shown to be the main decision-makers regarding the application of physical restraints. Nurses need to balance the ethical principle of beneficence through this practice, ensuring the safety of the patient, and the principle of autonomy of the person.

## 1. Introduction

The most comprehensive definition of restraint, resulting from a Delphy survey involving 48 international experts from 14 different countries, states: “Physical restraint is defined as any action or procedure that prevents free movement of the body to a chosen position and/or normal access to the body through the use of any device, material or apparatus attached to, or placed near, a person’s body and which cannot be easily controlled, or removed by the person” [1].

The available scientific literature offers different estimates of physical restraint use, depending on the country and the context examined. The study by Kruger and colleagues [2] considered 14 intensive care units of four hospitals in Germany, with a prevalence of the use of physical restraints ranging from 0% to 90%. Another multicenter study conducted by Benbenbishty and colleagues [3] in the intensive care units of nine European countries determined that the prevalence of the use of physical restraint varied from 0% to 100%. Although the findings of this study have to be seen in light of some limitations, due to the different number of samples included for each ICU, ranging from 15 to 319 patients per country, the results reflect the variability of restraint practice related to the peculiarities of each context considered. Physical restraint is commonly considered an accepted practice to ensure patient safety, with nurses playing a key role in the decision-making process [4,5,6]. The attitudes and beliefs of nursing staff appear to be decisive factors in the use of restraint [2].

The most frequently reported reasons for the use of physical restraint are the management of patient agitation and restlessness [3,7], prevention of falls and self-removal of life-support devices [3,7,8], such as: nasogastric tube, intracranial devices, endotracheal tube, central venous catheter and arterial catheters, peripheral venous catheters, surgical drains, urinary catheter, ECMO device catheters and tracheostomy cannula [9].

Regarding mechanical ventilation, De Jonghe and colleagues [10] collected the perspective of doctors in 121 intensive care units in France and 82% stated that more than 50% of patients undergoing mechanical ventilation had been physically restrained at least once. Another study conducted in Canada by Burry and colleagues [11] reported that 53% of 711 mechanically ventilated patients were physically restrained for an average of four days.

Facilitating factors for the use of restraint tend to be: (1) the clinical conditions of the patient, in terms of inability to cooperate with nurses or weaning from mechanical ventilation or spontaneous awakening trials, (2) overload, (3) lack of other alternative solutions [6].

In the study by Via-Clavero and colleagues [6], nurses defined the use of physical restraint to prevent self-removal of life support devices or to prevent falls, in restless or agitated patients, as beneficial.

Among the negative effects, nurses acknowledge physical and psychological patient harms associated with their use, particularly feelings of vulnerability and family and professional distress [6], although the patients often have little or no memory of their physical restraint experience. A recent review shows that communication between nurses, the patient and his/her family is the most crucial determinant of the level of anxiety for the patient and the family when using physical restraint. Anxiety tends to decrease when nurses communicate the reason for restraint and families feel reassured knowing that restraints prevent the patient from dislodging and/or removing devices [12].

Given the fact that, despite the worldwide promotion of a “restraint free” model of care due to the questionable ethical and legal issues and the many adverse physical and psychosocial effects of physical restraints, their use remains relatively high, especially in intensive care settings, it would be important to obtain a deeper level of understanding regarding nurses’ perceptions on physical restraints use, in order to better understand how nurses reason and act in their daily clinical practice. Therefore, the aim of the present study was to explore the experiences of nurses using physical restraint in the intensive care setting.

## 2. Materials and Methods

### 2.1. Study Design

A descriptive qualitative research design through semi-structured interviews was used [13]. The consolidated criteria for reporting qualitative research (COREQ) checklist were used to ensure quality reporting of this qualitative study.

### 2.2. Setting and Sampling

A purposive sample of 20 intensive care nurses was selected between December 2018 and February 2019. The study was conducted in 20 intensive care units (ICUs) (6 medical, 6 surgical, and 8 medical-surgical ICUs) of ten hospitals in Northern part of Italy (six public general hospitals and four university-affiliated hospitals). The units had from six to eighteen beds, and a nurse-to patient ratio ranging from 1:1 to 1:2. Half of the units had “restricted visiting policies” for an hour a day, the other half had “partly restricted policies” for six hours a day (from 2 p.m. to 8 p.m.).

Nurses have been included in the study on a voluntary basis and had to have the following characteristics: (1) have been working in an intensive care unit for at least 3 years, (2) providing consent to participate in the study. In order to maximize variability and enrich the data, the investigators selected nurses with different gender, age, professional and work tenure in ICUs, educational level, and working in different hospitals. Data saturation was reached at the 18th interview, and two more interviews were added to ensure the completeness of the data.

### 2.3. Data Collection

The data collection was carried out through the administration of face-to-face semi-structured interviews conducted in a dedicated setting at a time convenient to participants. Consent was obtained at the beginning of the interviews. The interviewers were nurses who were in their second year of the master’s degree in Nursing and Midwifery Sciences at the University of Verona and underwent interview training.

A total of twenty interviews were performed, with an average duration of 30 min. An interview guide was developed by the principal investigator and was pilot tested in the first two interviews, requesting minor revisions. These two interviews were included in the data analysis because of their relevance.

Each interview contained the lead questions listed in Table 1.

All interviews were conducted in Italian, audio-recorded and subsequently transcribed verbatim, by the same nurses who performed them.

### 2.4. Data Analysis

The interviews were analyzed using a qualitative descriptive approach by adopting the five-stage process of Sandelowski and Barroso (2002):(1)Familiarizing oneself with the material (holistic reading): each text/interview was read repeatedly in order to achieve an overall view of the material;(2)Identifying the significant text units: the significant words and/or phrases with respect to the research question;(3)Condensing each textual unit with a descriptive label: each significant part of the text is given a descriptive label;(4)Grouping the labels into categories: the labels were compared and those with a certain similarity were grouped together. Each group of similar labels constitutes a category;(5)Grouping the categories into themes: the categories produced were examined in order to collect them into homogeneous groups, each of which constitutes a category of a higher order (themes).

### 2.5. Trustworthiness

Various strategies were used to improve the quality of the present study. The transcripts were anonymized and representative quotations from the transcribed text were reported to support study’s findings. Data analysis was performed independently by a nurse with advanced education and working experience in intensive care and two experts in the field of qualitative research. Disagreements were resolved by discussion. Reflexivity was practiced minimizing the influence of preconceived perspectives and, prior to entering the research field, pre-existing knowledge, ideas and assumptions on physical restraint, were journaled.

## 3. Results

Of the 20 nurses selected by purposive sampling, 12 were female, while the average working age of the participants was 15 years. Thirteen nurses had post-basic education: three nurses had a master’s degree in nursing sciences, 10 nurses had a master’s degree in advanced clinical practice. Sixteen of the nurses interviewed had more than five years of previous working experience in intensive care and were considered “expert” nurses according to the classification of Benner and colleagues [14]. Sociodemographic data of interviewed nurses are presented in Table 2.

The results of the synthesis and processing of the qualitative interviews collected, grouped according to the themes identified in the process of their analysis, are presented below (Figure 1).

### 3.1. Definition of Physical Restraint

The interviewed nurses gave their definition of physical restraint, providing different meanings to this term.

Some of the interviewees referred to restraints as the tools they use in their routinely clinical practice, for example bed rails or cuffs. In this regard, a nurse stated: “*For me, restraint is the use of means, like devices called cuffs or anklets that are hospital provided tools, made of textile material. They include segments with softer sponges that are put around either the wrists or ankles of a patient to restrain his movement, adjusting them according to the size of the wrist or ankle*” (int.3).

Some of the nurses perceived restraints either as a positive or negative practice, giving two opposite perceptions of the phenomenon. Indeed, on one hand the ICU nurses considering restraints as a way of caring for the patients stated that they are “*means whose purpose is to ensure care*”. Under this perspective, nurses consider them as additional protection tools for the patient, whom otherwise may tend to remove essential medical devices, as it may be the endotracheal tube. “*I see it as a way to protect and care for the patient, even if it may seem to be something… like torture” … but it is not because it protects the patient*” (int.20). Therefore, the use of restraint it is felt as a necessary “*evil*” to control these adverse events.

On the other hand, restraint is lived and perceived by other healthcare professionals as a form of imprisonment, because of the action aiming at trying and stopping the patient, which in practice is not free to pursue his will and wishes: “*I think it is a prison for the patient*” (int.13).

### 3.2. Who Decides to Restrain?

From the collected interviews emerged that the decision to apply restraints to the patient comes mainly from the nurses. This circumstance is probably related to the fact that nurses are the health professionals spending more time with the patients, compared to other health professionals. A nurse specifically states this aspect: *“… because we are basically on the patient … I am the one who notices if the patient is going to tear the CVC rather than another device*” (int.1).

This decision-making process considers the risk-benefit ratio, which can be different according to the unique situation of each patient: *“It depends on the risk-benefit. In other words, an assessment is performed, and a decision is taken accordingly*” (int.5).

In case of urgency, the nurse, after an assessment based on his/her previous experiences and best clinical judgement, may decide autonomously to apply some restraints. The available restraining tools will be used with increasing gradualness: “*Usually, in emergency cases, I decide to start with the simplest restraints and then, if needed, I apply stronger measures after consulting my colleagues*” (int.8).

However, in ordinary situations, the nurse proposes to the physician the application of physical restraints. After the discussion of the case the team takes a shared decision, which, in most situations, confirms the proposal initially introduced by the nurse: “*There is often a discussion among colleagues before applying these devices, it is not a decision of the single health professional*” (int.17).

It is interesting to highlight that the physicians tend to endorse the nurse’s decision to apply physical restraint relaying on the relationship of trust that they have with the nurses.

### 3.3. Reasons behind the Restraint Use

Physically restraining a person to protect his/her safety is one of the most frequently emerging aspects in the collected interviews: “*I just see it as a safety measure for the patient. The decision of applying restraints, is not something we like to do, instead we use them just to prevent the patient from hurting himself, because 99% of patients are basically under sedation and they can unintentionally take off something vital*” (int.12).

Physical restraint is described as a practice routinely used as a preventive measure to avoid danger to the patient, as this nurse states: “*I approached her [patient] and explained that restraints were applied as a preventive measure so that the tube would not be removed during the sedation period*” (int.7). The main reasons concern the risk of self-removal by patients of the endotracheal tube and/or other devices following the strict recommendation of the physician to prevent unpleasant episodes: “*Sometimes they [physicians] say: I recommend that you look after the patient because it is your responsibility if patient removes the tube. This happens especially if one [patient] has had a difficult intubation, so it already gives you an idea of what to do*” (int.4).

A secondary aim is to prevent damage caused by the repositioning of a device that was unintentionally removed, because “*Intensive care patients, being critical patients, have a whole series of accesses and clinical accessories that are needed for monitoring. This is precisely the problem. When a device is accidentally dislodged, it’s not just that the patient takes away a device. In certain situations, those devices are essential and putting them back on does more damage, therefore preventive restraint prevents further damage*” (int.4).

Among the reasons that lead nurses to use restraints is the inability of the patient to receive direct assistance from a caregiver. Despite the fact that in isolated cases it has been reported that restraint is improperly used “*to adhere to a social norm*” (int.19) of the group, it is commonly used for the good and the protection of the patient’s safety, since the nurse cannot constantly monitor the patient: “[…] *either you stay there and keep him still for 8–10 h, which is impossible, or… hem… you have no other means*!” (int.19).

### 3.4. Physical Restraint Used as the Last Option

Physical restraint is considered by nurses to be the last option, to be adopted when all the alternative strategies previously attempted have not been effective in obtaining the patient’s cooperation: “*before arriving to restraint we try to put in place a whole series of maneuvers that may avoid this outcome*” (int. 2).

However, when the patient seems agitated and not very compliant, it is important for the nurse and the physician to share a discussion considering all the alternatives to physical restraints, like potential dose and medication adjustments to his/her therapy: “*Generally, before to take the decision to restrain, there is an exchange of opinions first of all with the doctor who could eventually decide a different pharmacological therapy to try to partially sedate the patient or anyway make him a little more compliant*” (int. 16).

Communication with the patient is the strategy most shared and applied by the interviewees to prevent the patient from harming himself. In this case, the nurse tries to establish a communicative relationship with the patient. The health professional firstly tries to calm the patient if he/she is showing signs of agitation, establishing a relationship of trust. “*For us [nurses] it is essential before arriving to restrain, have faith in the patient, especially if he/she is awake, and so to talk and to stay close to the patient*” (int.14). The results of this approach are described by a nurse as follows: “*At my explanation, the patient calmed down and I took off the cuffs to give her proof of trust*” (int. 15).

In particular, the nurses try to re-orientate confused and disorientated patients through communication: “*He calmed down, but I suppose that agitation represented a way of expression of sadness for not being able to communicate with us. As soon as he regained the ability to communicate, he was back to normal*” (int. 6). Among the useful strategies to prevent the use of physical restraint, remaining in the domain of communication, some nurses describe the importance of eye contact, the ability to read lips and non-verbal language.

Communication and closeness to those patients which are experiencing weaning from sedative drugs or mechanical ventilation are considered common and useful interventions: “*in the awakening period we stay closer to the patient, and this is a shared caring approach within our team* […]” (int. 10).

The nurses’ ability to show a positive and accommodating attitude toward the patient’s requests can make the difference between avoiding the use of physical restraints or not; with reference to this, it was highlighted in some interviews that mobilization can be a valid alternative, as testified by this nurse: “*It pains me to tie the patients, they can already only move their hands, they are full of wires, stuck in bed and unable to move on their own, …, you try with the elderly person maybe, …, you tell him “do you want to try to put yourself on the side? at that point you try to turn him and make him more comfortable, because he may simply have some difficulties to sleep due to the position and so you try to support him*” (int. 18).

Since the self-removal of devices is one of the main reasons for the use of restraints, some nurses point out that a winning strategy may be to hide these devices from the patient’s sight and to associate the simultaneous removal of restraints and devices: “*Many times, there is the risk that the intubated patients, which happen to not being under deep sedation and therefore a bit awake, obviously try to remove the tube, which is the thing that most bothers them anyway, as you would expect. For this reason, we try to hide the invasive device as best as we can, for example the nasogastric tube, maybe we make them go around behind the head so that it’s easier to forget that it exists, and the patient is less tempted to remove it*” (int. 9).

On the other hand, referring to the advantages of removing invasive devices, one nurse says: “*Quite simply, even removing a urinary catheter—a device that all our patients have—reduces agitation. By meeting the needs of the patient, we can reduce discomfort and improve the quality of care*” (int.7). Providing pain relief to the patient is mentioned by one nurse as another effective intervention in preventing the use of physical restraint.

In the Italian context, usually the relatives are involved in the process of vigilance at the patient’s bed, and in these moments the restraints are usually removed, as testified by this nurse: “*When the relatives were present, we also tried to avoid the restraints, explaining to the relatives why we needed to maintain in place the invasive devices and trying to involve them in the daily activities*” (int. 11). In some cases, the minority of the cases according to the nurses, there are situations in which the presence of family members leads to a worsening of the patient’s agitation, so the nurse needs to find ways to calm the patient and mitigate this reaction. The presence of the relatives at the patient’s bedside may help to prevent the placement of restraints and the removal of invasive medical devices, however in isolated cases “*It was the relatives who told us: tie him up because he will tear everything down*” (int. 4).

From the interviews emerged that a nurse-to-patient ratio of 1:2 makes it possible to limit the use of physical restraints, as this nurse asserts: “*Fortunately we tend to use not as many restraining methods compared to other contexts, because our nurse-to-patient ratio is one to two*” (int. 12). A key strategy to avoid the application of restraints may be to ensure a 1:1 nurse to patient ratio in case of highly agitated patients: “*If we have available beds, we try to have a one-to-one nurse-to-patient ratio for confused and agitated patients, in doing so we often avoid having to apply restraining methods*” (int.17).

### 3.5. Family Involvement

Physical restraint has a psychological impact on the family members of ICU patients, but if it is supported by effective communications with the nursing staff, the family normally understands and supports it: “*The relatives understand that what you are doing is for the protection of their loved one, …, so in my opinion the initial approach is pretty difficult, but after a proper conversation where we explain why we have taken that decision, it is well received“* (int. 5).

Communication is important to reduce the psychological impact of intensive care on family members, who feel helpless seeing and trying to cope with their loved one in an ICU bed, as the nurse says: “*For a family member entering the ICU to see his/her loved one tied up is a strong impact, and to prepare them we use all our communication skills, it is one of the first things we tell them when family members come in*” (int. 20).

In order to help the family cope with the impact of the restraining methods, the nurses highlighted how appropriate listening skills are important: “*In my opinion, you have to know how to listen and contribute to the relationship also choosing the appropriate words*” (int. 1). At the same time, empathy is also part of nursing and it helps to understand what the family members of the patient are going through: “*In that moment I shared their state of mind and their feelings. I agreed with them*” (int. 3).

Despite the difficulties in accepting the restraints, sometimes the acknowledgement of the necessity of their use leads the relatives to further ask for their application when they feel that the patient’s agitation risks undermine his safety, showing trust in the healthcare staff: “*There are relatives who, however, when they see their family member in a state of mental disorders and they know that I previously had to tie him up, they tell you “at night, I recommend you tie him up, because if not he will hurt himself*” (int. 10).

### 3.6. Nurses’ Feelings about Restraint

Nurses experience a variety of feelings when they have to apply physical restraints to their patients. For some of them, the habit of restraint and the state of necessity that can justify its use leads them not to develop many reflections on it. This thinking process pushes the nurse to live restraints as a common and routine practice: “*I did not have particular sensations also because the method is so common that I do not reflect so much about it. Maybe because you are so convinced that it is useful to avoid a possible danger you don’t ask yourself big questions like: Should have I done it differently? I do not have any particular feelings to express*” (int. 7).

In this case the nurses feel “*professionally peaceful*” because it is part of their job. They feel relieved for the safety of the patient and the prevention of the removal of invasive devices. Some nurses consider the use of restraints unpleasant even though it is sometimes a necessary practice: “*It is unpleasant to apply restraints, but it is also necessary*” (int. 18).

On the other hand, most of the interviewed nurses, when asked about the dignity of this practice, identified themselves with the patient and described the following feelings:

-anxiety: “*And it becomes difficult to manage, because to me it creates a lot of anxiety, because I am really afraid, they will hurt themselves, they will fall off the bed, they will tear what they should not take off […]*” (int. 2);

-pity: “*Not well because in his moments of lucidity, it is bad to use the word pity, but it makes you feel sorry, in the sense that you see a person with whom you could potentially have a relationship*” (int. 13);

-frustration: “*When I adopt such an important technique of restraint… I feel almost a sense of frustration for the patient himself*” (int. 18);

-compassion: “*I always live it badly, in the sense that it makes me sad because I always think if there was a relative of mine, if I were to see him like that I would really feel sad thinking about him. It is not easy to think that he may spend the last moments of his life tied up like a salami, but unfortunately our reality is difficult*” (int.17).

In the context of intensive care, the nurses report that in many situations the implementation of physical restraint “*is the only way*” (int. 3). It is in fact considered in many cases as the last and only option available to preserve the safety of the patient.

## 4. Discussion

Several facets of the definition of physical restraint emerge from this study. Just as in the study by Langley and colleagues [4], physical restraint could be defined as a “balancing act”.

From the analysis of the interviews collected, it appears that among the nurses coexist two opposite perceptions of physical restraints: the positive view considers restraints as necessary means to ensure the safety of the patient, weighting the safety of the patients as an outcome of primary importance compared to the right of patients to move autonomously. On the other hand, other nurses maintain a negative perception of restraints, described through the words “tie” and “block”. In this case some nurses arrive to define it as a form of “captivity”.

Indeed, according to other studies [5,15], although ensuring patient safety is the most used justification for the use of physical restraint, nurses may face uncertainty and conflict in choosing between patient safety and autonomy [4,16].

Nurses have been considering the use of physical restraint as a common intervention and they have been playing a key role in the decision-making process to initiate the use of restraint [5,17]. Many times, this assessment and subsequent decision-making is taken by nurses, justifying the decision as a state of necessity [18]. In other situations, the nurse proposes the use of restraints to the physician, who confirms the nurses’ request since the relationship of trust with the nurse [18]. Among the reasons given by nurses to justify the use of restraints in ICU, there was patient agitation [5,6,18,19,20] the patient being in danger to himself and/or others [19] and being uncooperative [6]. The dangerousness of the patient’s agitated state has to be considered in relation to the risk of self-removal of invasive devices [5,6,19]. In this regard, the device that raises more concerns among nurses due to the dangers of self-removal is the endotracheal tube [3,6,7,19,21]

From the interviews it emerges that in addition to the damage caused by the removal of invasive devices, there are also injuries caused by the repositioning of these devices, which are necessary to support vital functions. On the other hand, collaborating patients who are compliant with the therapy administration and with the healthcare staff are not contained, as stated by Via-Clavero and colleagues [6].

Even though the choice to resort to physical restraint is considered as the last option after the failure of other interventions, it emerged from some nurses that restraint is a practice handed down from experienced nurses to novices.

The strategies mentioned in this study by the nurses as potentially useful interventions in order to avoid physical restraints place in the foreground an effective communication with the patient, in an attempt to re-orientate him when disoriented and to regain communication skills, establishing a relationship of trust with the patient, as also supported by the study of Evans and colleagues [22].

The proximity of the nurse to the patient in the process of weaning from sedation or mechanical ventilation reveals another practical declination of the relational aspect with the patient. By being accommodating to the patient and providing interventions aimed at taking care of comfort [18], the nurse can gain the patient’s trust, being able to calm the patient and in some cases to completely resolve his/her state of agitation.

As already pointed out, the use of physical restraint and the presence of invasive devices are closely related. The decision to hide these devices from the patient’s sight and to promote their early removal are considered effective interventions by the interviewed nurses.

An aspect of great importance that emerges from the interviews is the presence of adequate analgesic coverage to the patient, since that untreated pain may be a great source of restlessness for the patient [22].

Some studies state that interprofessional collaboration is a key determinant to avoid the need to resort to physical restraints [6].

The family is an important resource for nurses. This is even more so following the introduction of “open intensive care”. Indeed, this mode of intensive care management finds at its core the active involvement of the family in the process of care of their relatives [6,18]. Following this approach, the nurse takes care not only of the patient but also of his or her relatives. The study by Kandeel & Attia [18] showed that the majority of the nurses stated that neither the physically restrained patients (70.6%) nor their families (68.6%) received any education about the reasons leading to the use of physical restraint. Once again, communication becomes a cardinal element, which needs to be clear and effective both with the patients and with their family members. An essential element that determines the effectiveness of communication is the ability to listen and to be empathetic. Patient anxiety is likely to decrease when nurses communicate the reason for restraint, and family members tend to be reassured in knowing that the patient is restrained for his safety, due to the risk of pulling and therefore removing essential medical devices.

Furthermore, nurses emphasize that an adequate nurses-to-patient ratio in ICU, which can be 1:2, may allow nurses to increase the time spent at the patient’s bedside and in cases of particularly agitated patients, ensuring a 1:1 ratio can be an effective strategy to avoid the use of physical restraints.

The nurses interviewed show different feelings and emotions related to the act of restraint. If on the one hand they say they are reassured and relieved to have ensured the safety of the patient. On the other hand, by identifying themselves in the patient, they experience negative feelings, including anxiety, pain, frustration, and tenderness. Despite the fact that this practice is not pleasant, and given the ambivalence of the feelings experienced by nurses, it emerges that in the intensive care setting this practice is still seen as necessary and often as the only option available for the protection of the patient [23].

### Limitations

This study has some limitations as it was conducted using a qualitative approach, there is the potential for the researcher’s experiences, norms and values relating to the phenomenon to influence findings of the study. To minimize the interviewer’s personal perceptions many efforts has been taken as described in trustworthiness section. Another limitation was the relatively small sample size; however, data saturation was reached and the study provides useful insight into nurse’s experiences of physical restraint use. Due to the sensitivity of the topic, our participants might have avoided sharing some aspects of their experiences. Of course, we attempted to manage this limitation by establishing a trustful relationship with them. The study is limited to the Northern Italian context, thus transferability of these results to other contexts may be challenging, even if the qualitative approach does not aim to generalize findings, and it is necessary to consider nursing care in the intensive care units as a process with the same goal all over the world.

## 5. Conclusions

Talking about physical restraint evokes different thoughts and feelings. Nurses, which are the professionals most present at the patient’s bedside, have been shown to be the main decision-makers regarding the application of physical restraint. Nurses need to balance the ethical principle of beneficence through this practice, ensuring the safety of the patient, and the principle of autonomy of the person. Among the words of the nurses collected through this study, “communication” is one of the most emphatically repeated. Effective communication is a fundamental approach to attempt with the patient to prevent the application of physical restraints and to understand and accept their possible use. Nurses often consider the use of physical restraint when they do not see any possible alternative solution. It is therefore necessary to assess whether they are willing to accept the occurrence of events such as self-extubation of the patient or the risks associated with the removal of invasive devices and the consequences that could result, thus avoiding the use of devices that limit the autonomy and freedom of the sick person.

Nurses themselves and nursing managers can benefit from these findings to gain an overview of the behaviors of nurses in applying physical restraint in the intensive care setting. These insights highlight the importance of encouraging education, training, and policy development on physical restraint practices. Even if nurses are the main decision makers of physical restraint use, physicians, patients, and family caregivers should also be considered when designing and developing interventions to minimize the use of restraint.

## Figures and Tables

**Figure 1 ijerph-18-09646-f001:**
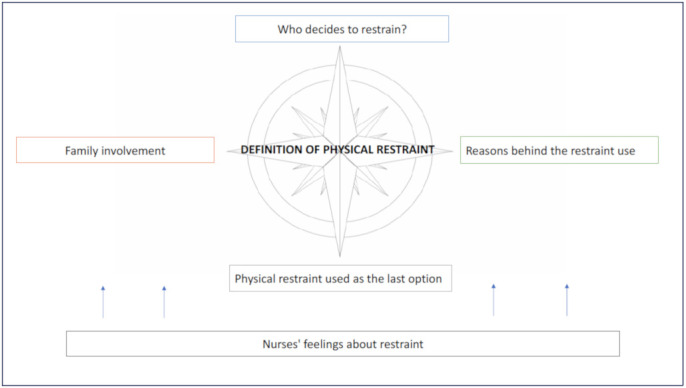
Overview of the emerged themes from nurses’ point of view on physical restraint use.

**Table 1 ijerph-18-09646-t001:** Questions of semi-structured interview guide.

Question Order	Questions of Semi-Structured Interview Guide
A	If you think about the concept of physical restraint, what comes to your mind?
B	Could you please describe an episode/situation in which you decided to use physical restraint, for example in the last week? Possible stimulus questions: What was the triggering event? How did the decision to restrain the patient take place? Was an individual decision or shared with other healthcare professionals? Were alternative strategies to physical restraint considered?
C	What impact do you think physical restraint had on the patient? And on the patient’s caregiver/s?
D	What was the impact on you when you experienced the patient be re-strained?

**Table 2 ijerph-18-09646-t002:** Sociodemographic data of interviewed nurses.

Demographics		Number of Participants
Gender	Female	12
	Male	8
Education level	Bachelor’s degree in Nursing	7
	Master’s Degree in Nursing Science	3
	Master’s Degree in Advanced Clinical Practice	10
Age (years), mean (SD)		37.8 (7.9)
Professional tenure (years), mean (SD)		15 (7.2)
Working experience in intensive care setting	0–5 years	4
	6–10 years	7
	11–20 years	6
	>21 years	3

## Data Availability

Data are stored at the Department of Diagnostics and Public Health, University of Verona.

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
