# Peer review of "Nurses’ Views on the Use of Physical Restraints in Intensive Care: A Qualitative Study"

_ijerph, 2021, doi:10.3390/ijerph18189646_

Round 1
Reviewer 1 Report
I enjoyed reviewing this interesting, informative and insightful manuscript. However, there are some writing/editing issues that the authors should consider and address.
In the Introduction,
line 41, "...Germany with a prevalence of the use of physical ...".
Line 45, "...study were important due to the different number ...".
Line 56, Regarding mechanical ventilation, DeJonghe ...".
Line 64, "weaning from mechanical ventilation or ...".
Line 73, "...the level of anxiety for the patient and family ...".
Line 80, "...care setting. A deeper understanding of the nurses' perceptions on ...".
In the Material and Methods section,
line 93, "1) working in an intensive care unit for ...".
Line 94, "...and enrich the data, the investigators selected ...".
Line 95, "...and work tenure in ICUs, educational level, ...".
Line 138, "search experts in the field of qualitative research."
In the Results section,
line 174, "..."I think it is a prison for the ...".
Lines 174-175, "...to contain the patient involves in most cases the nurses, as they are ...".
Line 176, "patients, as this nurse points out:"...because ...".
Line 181, "In case of an emergency the nurse, guided by ...".
Lines 221 & 222, "...last resource intervention, to be adopted when all the alternative ...".
Line 281, "...ratio of 1:2 makes it possible to limit the use of ...".
Line 299, "...of the restraining methods, the nurses highlighted how ...".
Line 307, "...patient's agitation risks undermine his safety, showing ...".
Lines 313 & 314, "...state of necessity can justify its use and leads them not to develop ...".
Line 319, "...it is part of their job. They feel relieved ...".
In the Discussion section,
line 343, "...restraint could be defined as a ...".
Line 359, "interviews, which emerged how nurses considered ...".
Line 363, "of restraint [5]. In fact, it does not come as a surprise ...".
Line 364, "...and managers of restraints [17]."
Line 369, "[5, 6, 18-20], the patient being in danger to himself and/or ...".
Line 399, "...supported by the study of Evans D et al. [22]."
Lines 406 & 407, "...majority of the nurses stated that neither the ...".
Line 421, "safety of the patient. On the other hand, by identifying ...".
Author Response
Dear Reviewer, according to you suggestion, we re-wrote all the above cited statements.
Reviewer 2 Report
Dear authors,
I read the article entitled: Nurses' views on the use of physical restraints in intensive care: a qualitative study.
The main message is that "physical restraints are the two faces of a coin".
The article is interesting and falls in an area of interest for ICU nurses and physicians. The manuscript is written sufficiently.
However, please do not use contraction form in medical literature example:
page 5, line 211 "it's" but use it is and so on.
- Abstract and M&M: please improve the methods;
- Introduction -ok
- Methods - Please better-specified the setting.
- Discussion Would you please try to shorten it by 25%? Limitation - Explain better your study limitation.
- Conclusions - No problem
- References -ok
- Finally, a paper without a figure is difficult to read and understand, and please enrich this manuscript with tables and figures. Best Regards
Author Response
Dear authors,
I read the article entitled: Nurses' views on the use of physical restraints in intensive care: a qualitative study.
The main message is that "physical restraints are the two faces of a coin".
The article is interesting and falls in an area of interest for ICU nurses and physicians. The manuscript is written sufficiently.
However, please do not use contraction form in medical literature example:
page 5, line 211 "it's" but use it is and so on.
We revised the manuscript according to your suggestion.
Abstract and M&M: please improve the methods;
We added some more information about the methods in the abstract.
Introduction -ok
Methods - Please better-specified the setting.
According to your suggestion, we added more details about the study setting in the methods section.
Discussion Would you please try to shorten it by 25%? Limitation - Explain better your study limitation.
We added a specific paragraph to describe study limitations.
Conclusions - No problem
References -ok
Finally, a paper without a figure is difficult to read and understand, and please enrich this manuscript with tables and figures. Best Regards
This is an interesting point, we enriched the manuscript with two tables and one figure.
Reviewer 3 Report
Thank you for the opportunity to read the results of your research.
It is an important voice in the professional approach to the patient.
I like the thorough approach to the study, planning it according to ethical and methodological requirements.
I have some concerns about the manuscript presented:
1 - It seems to me that presenting data as percentages with such a small sample is not correct. It is better to give real values, they are more reliable.
2 - When quoting respondents' statements it would be good to give their number/respondent code in brackets, it makes the statement more credible.
3 - I miss indications concerning practical use of obtained conclusions.
4 - There are no indicated limitations of the study.
Author Response

(The authors gave the same response as above.)
